# Structural basis of the molecular ruler mechanism of a bacterial glycosyltransferase

Ana S. Ramírez [1], Jérémy Boilevin[2], Ahmad Reza Mehdipour[3], Gerhard Hummer[3,4], Tamis Darbre[2], Jean-Louis Reymond [2] & Kaspar P. Locher [1]

The membrane-associated, processive and retaining glycosyltransferase PglH from *Campylobacter jejuni* is part of the biosynthetic pathway of the lipid-linked oligosaccharide (LLO) that serves as the glycan donor in bacterial protein N-glycosylation. Using an unknown counting mechanism, PglH catalyzes the transfer of exactly three α1,4 N-acetylgalactosamine (GalNAc) units to the growing LLO precursor, GalNAc-α1,4-GalNAc-α1,3-Bac-α1-PP-undecaprenyl. Here, we present crystal structures of PglH in three distinct states, including a binary complex with UDP-GalNAc and two ternary complexes containing a chemo-enzymatically generated LLO analog and either UDP or synthetic, nonhydrolyzable UDP-$CH_2$-GalNAc. PglH contains an amphipathic helix ("ruler helix") that has a dual role of facilitating membrane attachment and glycan counting. The ruler helix contains three positively charged side chains that can bind the pyrophosphate group of the LLO substrate and thus limit the addition of GalNAc units to three. These results, combined with molecular dynamics simulations, provide the mechanism of glycan counting by PglH.

[1] Institute of Molecular Biology and Biophysics Eidgenössische Technische Hochschule (ETH), CH-8093 Zürich, Switzerland. [2] Department of Chemistry and Biochemistry, University of Berne, CH-3012 Berne, Switzerland. [3] Department of Theoretical Biophysics, Max Planck Institute of Biophysics, DE-60438 Frankfurt, Germany. [4] Institute of Biophysics, Goethe University, DE-60438 Frankfurt, Germany. Correspondence and requests for materials should be addressed to K.P.L. (email: locher@mol.biol.ethz.ch)

Protein N-glycosylation requires a lipid-linked oligo-saccharide (LLO) that acts as a glycan donor of the transfer reaction. In the Gram-negative bacterium *Campylobacter jejuni*, the relevant LLO is synthesized in the cytoplasmic side of the plasma membrane, in a process that involves several glyco-syltransferases that use uridine diphosphate (UDP)-activated sugars as substrates[1, 2] (Supplementary Fig. 1). The first step of LLO biosynthesis is the conversion of UDP-GlcNAc to UDP-N, N'-2,4-diacetylbacillosamine (UDP-Bac) by the enzymes PglF, PglE, and PglD[2, 3]. Bac-phosphate is then transferred to the lipid carrier undecaprenyl phosphate (Und-P) by PglC, yielding Und-PP-Bac[4, 5]. Subsequently, five terminal N-acetylgalactosamine (GalNAc) units are transferred by the enzymes PglA, PglJ, and PglH. Whereas PglA and PglJ each catalyzes a single GalNAc transfer reaction[6] (yielding α-1,3 and α-1,4 linkages, respectively), PglH catalyzes the addition of three α-1,4 GalNAc units[2, 7] (Fig. 1a). As a final step, PglI adds a glucose moiety, which leads to a branched β-1,3-linkage. The mature LLO is subsequently flipped to the periplasm by the ATP-binding cassette transporter PglK[8] and used by the single-subunit oligosaccharyltransferase PglB[9] for glycan transfer onto acceptor proteins.

Among the various glycosyltransferases (GTs) involved in this process, PglH stands out because it is a membrane-associated enzyme that catalyzes a well-controlled, processive reaction[2, 7]. However, in the absence of a high-resolution structure, it is not known how this B-family GT achieves its highly precise glycan length control. There are several other processive, length-controlling GTs involved in biological processes: For example, Alg11 catalyzes the addition of two α1,2-linked mannoses to Dol-PP-GlcNAc$_2$Man$_3$, during the biosynthesis of the eukaryotic LLO[10]. Another example is the mycobacterial galactofuranosyl transferase 1 (Glft1) that is involved in the biosynthesis of ara-binogalactan[11]. To understand the structural basis of such a counting mechanism, we combine structural studies and mole-cular dynamics simulations with synthetic, chemo-enzymatic, and enzymological studies, and uncover a mechanism by which PglH controls the number of successive glycan transfer reactions using a structural element that not only attaches the protein to the membrane surface, but also contains residues that interact with the pyrophosphate group of the LLO during elongation of the glycan moiety.

## Results

**Functional analysis and inhibitor trapping of PglH.** Although *C. jejuni* PglH is not an integral membrane protein, non-ionic detergents (Triton X-100 and dodecyl maltoside) were required for its successful purification in homogeneous and active form (Supplementary Fig. 2). Whereas the donor substrate UDP-GalNAc is commercially available, the LLO substrate (acceptor) is not. We therefore pursued a dual approach: for in vitro assays, the tri-saccharide-containing LLO Und-PP-Bac-GalNAc$_2$ (Tri-LLO) was extracted and partially purified from *Escherichia coli* cells expressing the *C. jejuni pgl* operon but containing inactivated *pglB* and *pglH* genes[12]. It was then incubated with purified PglH and UDP-GalNAc. The resulting LLO products were then used as donor substrates and their glycan moieties transferred onto a fluorescently labeled peptide using the purified bacterial oligo-saccharyltransferase PglB, followed by tricine sodium dodecyl sulfate–polyacrylamide gel electrophoresis (SDS-PAGE) analysis[13, 14] (Fig. 1b). While transfer of three GalNAc residues was very efficiently completed, we observed that PglH could add more than three GalNAc units to the LLO when the reaction was incubated for 16 h (Fig. 1b). Turnover rates of the PglH-catalyzed reaction could be determined by quantitating the amount of Und-PP-Bac-GalNAc$_5$ (Hexa-LLO) produced.

Even though Und-PP-Bac-GalNAc$_2$ could be efficiently extracted and used for in vitro activity assays, it requires 1% Triton X-100 for its solubilization and could only be concentrated to ~300 μM, making it impractical for co-crystallization experi-ments. We therefore devised a chemo-enzymatic approach starting from synthetic NerylNeryl-PP-GlcNAc[15, 16], which was extended to a tri-saccharide-containing LLO analog using recombinantly expressed and purified PglA and PglJ proteins[3] (Fig. 1c). Despite the fact that GlcNAc rather than N,N'-diacetilbacillosamine was present as the reducing-end sugars, all Pgl enzymes accepted the LLO analog as a substrate. The products of the PglA- and PglJ-catalyzed reactions were analyzed by Tricine SDS-PAGE after the glycans were transferred onto fluorescently labeled peptides using PglB[16] (Supplementary Fig. 3). The chemo-enzymatically generated trisaccharide-LLO analog was accepted as a substrate by PglH, but the turnover rate was 50-fold lower than that of the Und-containing LLO substrate.

In order to trap PglH in an intermediate state that could mimic the transition state of the reaction catalyzed, we synthesized the compound UDP-CH$_2$-GalNAc, which carries a phosphonate group at the anomeric carbon and is thus nonhydrolyzable (Supplementary Fig. 4). This compound could indeed inhibit PglH (half-maximal inhibitory concentration (IC$_{50}$) of $303 \pm 24$ μM) and exhibited a slightly higher affinity for PglH than UDP (IC$_{50}$ of $1023 \pm 87$ μM, Fig. 1d). Both the phosphonate compound and UDP could be used to trap ternary complexes.

**Architecture of PglH and donor substrate binding.** We deter-mined the structure of PglH in three distinct states. The structure of a binary complex with the donor substrate UDP-GalNAc was determined at 2.3 Å resolution, whereas the structures of ternary complexes including synthetic acceptor LLO and either UDP or UDP-CH$_2$-GalNAc were determined at 2.8 and 3.3 Å, respec-tively. Crystallographic data and refinement statistics are pre-sented in Supplementary Table 1. PglH exhibits the protein fold described for GT-B GTs, consisting of two Rossman-like domains connected by a linker region (Fig. 2a, Supplementary Fig. 5). The conformation of PglH is similar in all three structures, although a slight rotation of the C-terminal vs. the N-terminal domain is observed in the ternary complex, where PglH is bound to UDP and LLO analogs. The electrostatic surface potential of PglH shows that the alpha helix α2 is strongly amphipathic, con-tributing to the formation of a positive charged cluster and to a hydrophobic surface patch (Fig. 2b). Whereas similar helices were observed in other membrane-associated GTs such as the phos-phatidyl mannosyltransferase PimA[17–19], α2 is considerably longer in PglH than in other GT-B family members of known structure (Supplementary Fig. 6). This helix is termed "ruler helix" because it acts as a molecular ruler (see below).

The structure of PglH bound to UDP-GalNAc revealed the molecular details of the interactions with the donor substrate. The uridine moiety is positioned in a pocket formed by α12 and the loop connecting α10 and α11. There are two hydrogen bonds of the uracil moiety with the main chain of V247. The 2' and 3' hydroxyl groups of the ribose moiety form hydrogen bonds with E275, which is part of the conserved EX$_7$E motif along with E267, which interacts with the N-acetyl group of GalNAc (Fig. 3a). When we individually mutated E267 and E275 to alanine, we observed an ~300-fold decrease in the turnover rate of PglH compared to wild type (Table 1), in agreement with earlier observations[7]. The pyrophosphate moiety interacts via hydrogen bonds with R191, K196, and T271 (Fig. 3a). R191 and K196 are conserved in the GT-B family and were proposed to stabilize the leaving group (UDP) following the transfer reaction[20]. In the structures with LLO bound, R191 also interacts with the N-acetyl

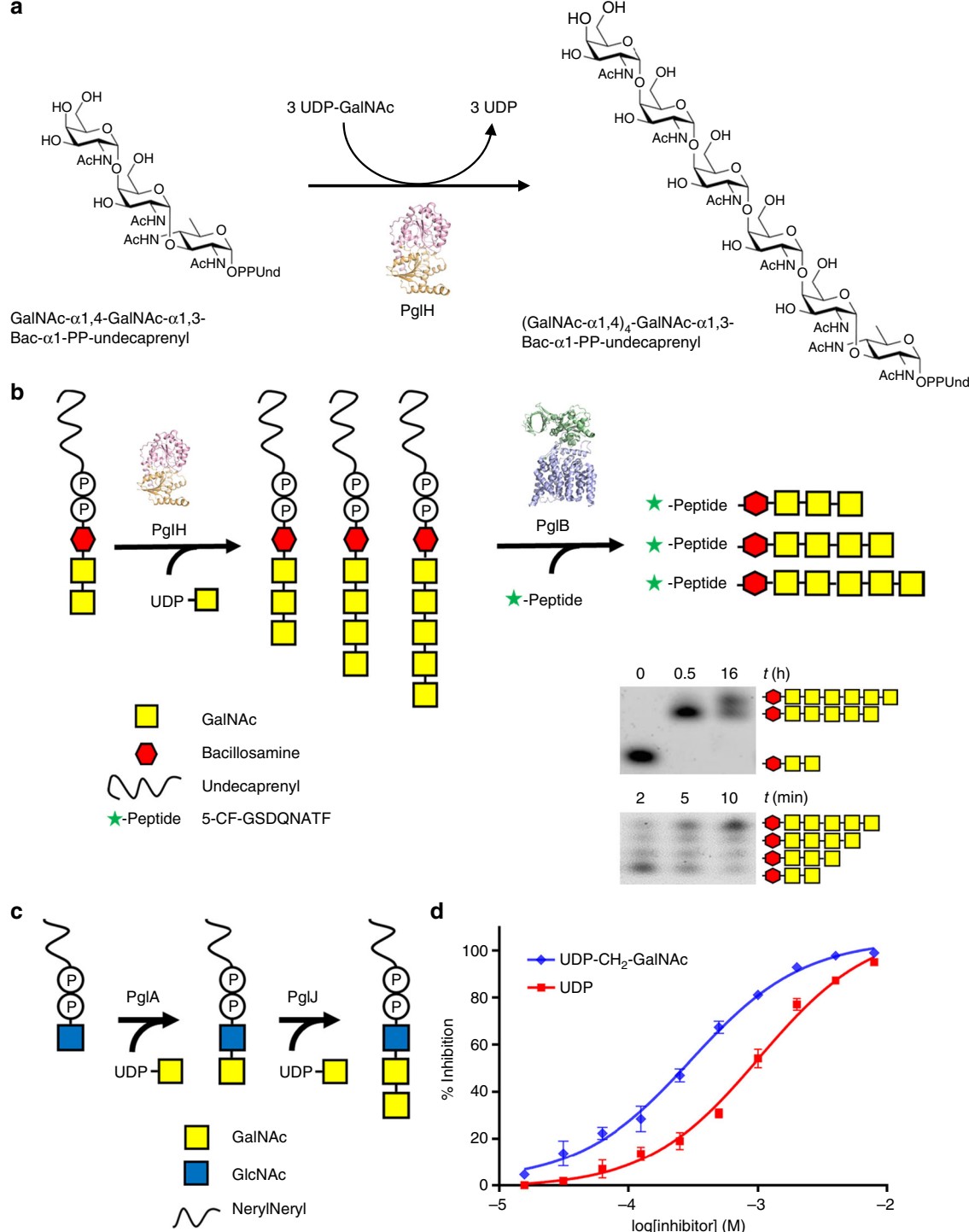

**Fig. 1** In vitro activity. **a** Reaction catalyzed by PglH during bacterial LLO biosynthesis. **b** Schematic representation of the in vitro assay used to quantitate PglH activity, with symbol legend at the bottom left of the panel. In a first step, tri-LLO is elongated by PglH. In a second step, the resulting glycan is transferred to a fluorescently labeled acceptor peptide using purified PglB protein. Green stars represent carboxyfluorescein attached to the N terminus of the synthetic peptide. **c** Schematic representation of the chemo-enzymatic generation GalNAc-α1-4-GalNAc-α1-4-GlcNAc-α1-PP-NerylNeryl, using purified PglA and PglJ proteins, with symbol legend at the bottom left of the panel. **d** PglH activity was measured in the presence of different concentrations of reaction inhibitors UDP or UDP-CH$_2$-GalNAc. Relative inhibition values were calculated, allowing IC$_{50}$ determination. Data points reflect the mean of three separate measurements, error bars indicate s.d.

group of the acceptor GalNAc (Fig. 3a, b). Individual mutations of R191 or K196 to alanine led to a complete loss of the activity, whereas the mutant T271A exhibited only a fivefold reduction in turnover rate compared to wild type (Table 1), suggesting that this side chain is involved in phosphate binding but does not have a catalytic role unlike R191 and K196.

The GalNAc unit is located in the cleft formed between the N- and C-terminal domains. Unlike in some structural studies of

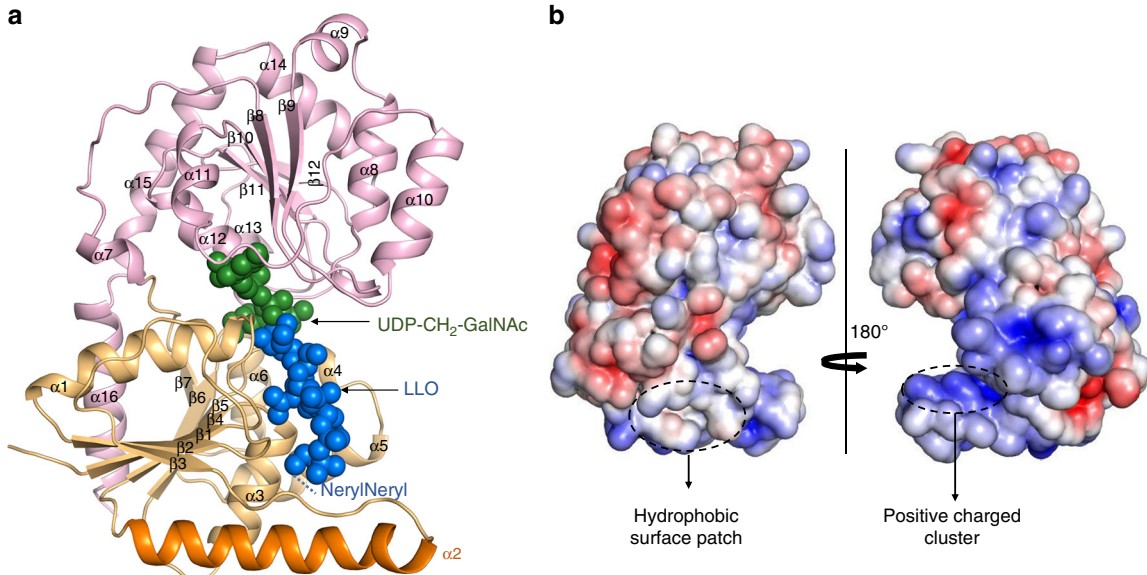

**Fig. 2** Crystal structure of PglH. **a** Ribbon diagram of PglH in complex with synthetic LLO and UDP-CH$_2$-GalNAc. N-terminal and C-terminal domains are colored in light pink and light orange, respectively. The N-terminal domain of PglH (residues 1 to 168) contains a parallel, 7-stranded β-sheet connected by six α-helices. The C-terminal domain (residues 185 to 357) contains a parallel, 5-stranded β-sheet connected by seven α-helices and a terminal α-helix that folds back to the N-terminal domain. Alpha helix α2 (termed "ruler helix") is shown in dark orange. Bound substrates are shown as spheres (green for UDP-CH$_2$-GalNAc, blue for LLO analog). **b** APBS-calculated electrostatic surface potential of PglH, color coded from blue (most positive) to red (most negative) to white (uncharged). The orientation of the left panel is identical to that in **a**

GT-B enzymes[19, 21, 22], the saccharide moieties were clearly visible not only for the non-hydrolyzable UDP-CH$_2$-GalNAc, but also for the functional UDP-GalNAc (Supplementary Fig. 7). The 6' hydroxyl of GalNAc forms a hydrogen bond with the side chain of H118 (Fig. 3a). Mutation of H118 to alanine led to a decrease of about 10-fold in the turnover rate (Table 1), supporting the stabilizing role of this interaction, which was also observed in the structures of other retaining GTs including MalP[23] and OtsA[24]. The structure can explain the specificity of PglH for GalNAc over GlcNAc: whereas the axial hydroxyl group at C4 of GalNAc can be accommodated by the enzyme, an equatorial hydroxyl group (present in GlcNAc) would generate a steric clash with the main chain carbonyl oxygen of L269 (Supplementary Fig. 8). The neighboring residue P270 may contribute to the rigidity of this region and therefore help in preventing the binding of UDP-GlcNAc. We observed that PglH did not accept UDP-GlcNAc as a substrate even after overnight incubation of the reaction mix, whereas the reaction with UDP-GalNAc under the same conditions was complete after 20 min (Supplementary Fig. 8).

**LLO binding site and geometry of the active site**. In both ternary complex structures, we could observe density for the three saccharides and the pyrophosphate group of the LLO analog, whereas the lipid tail was disordered (Supplementary Fig. 7). The LLO binding site is formed by the N-terminal domain of PglH and reaches from the catalytic pocket to the ruler helix α2. While there are distinct interactions with the terminal GalNAc residue, PglH provides only limited contacts to the reducing-end GlcNAc and the central GalNAc, in agreement with the finding that there is no strict requirement for a bacillosamine at the reducing-end position. The structure of PglH bound to LLO and UDP-CH$_2$-GalNAc visualizes an intermediate before reaching the transition state, whereas that bound to UDP can mimic a state after the transfer has been completed. In the

ternary complex with UDP-CH$_2$-GalNAc, the GalNAc moiety is in a "bent-back" conformation towards UDP, similar to what we observed in the structure of the binary complex of PglH with UDP-GalNAc and to what was observed in other GTs including PimA[25], MshA[26], and HepE[27]. This conformation allows the C4 hydroxyl of the terminal GalNAc of the LLO to be positioned at a distance of ~2.7 Å from the anomeric carbon of UDP-CH$_2$-GalNAc and ~3.4 Å from the methylene group mimicking O4 of UDP-GalNAc. In addition, it formed a hydrogen bond with the oxygen from the β-phosphate of the donor substrate analog (Fig. 3b, c). The orientation of the terminal GalNAc of the LLO is locked by two interactions of the C2-acetamido group with side chains of R191 and E18 (Fig. 3b, c). Mutation of these two residues to alanine abolished PglH activity (Table 1). However, because R191 also interacts with the pyrophosphate group of the leaving UDP, the R191A mutation has a dual effect. The absence of a potential nucleophile at the β-face of the anomeric carbon of UDP-CH$_2$-GalNAc might suggest that PglH uses a S$_N$1-like mechanism. The lack of evidence of the formation of an intermediate covalently bound to the enzyme was used to argue that most retaining GT-B members do not use a double displacement mechanism[20, 28].

**Counting mechanism**. The ternary complex structures reveal that the pyrophosphate group of the tri-LLO analog is appropriately positioned to form salt bridges with the positively charged residues K75 and R72 located in the ruler helix, whose side chains point towards the bound substrate (Fig. 4a). In addition to K75 and R72, there is a third positively charged residue in the ruler helix, K68, located one helix turn apart from R72. Given that PglH processes LLOs with three, four, or five saccharide moieties, we hypothesized that with every addition of a GalNAc moiety, the pyrophosphate group gets pushed further from the active site, successively interacting with K75, R72, and K68. We therefore analyzed the effect of mutating these residues

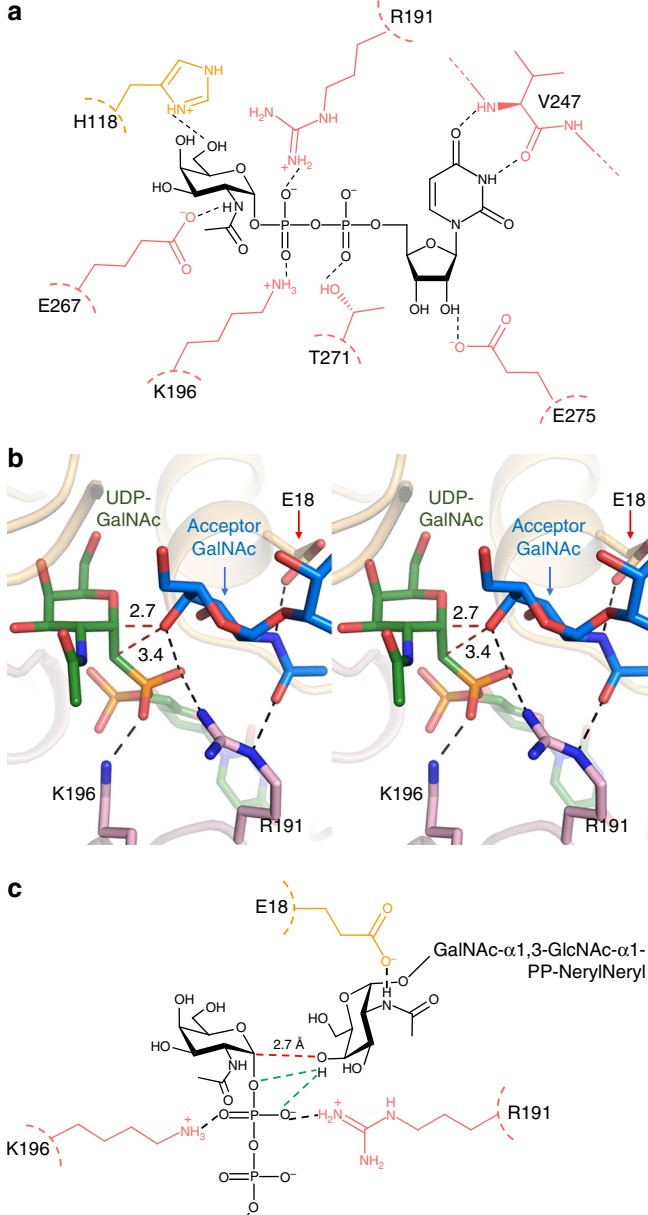

**Table 1 Turnover rates for PglH mutants**

| PglH | Turnover rate (Hexa-LLO·PglH$^{-1}$ s$^{-1}$) |
|---|---|
| WT | $2.00 \pm 0.34$ |
| H118A | $0.19 \pm 0.03$ |
| T271A | $0.34 \pm 0.08$ |
| E267A | $0.008 \pm 0.001$ |
| E275A | $0.006 \pm 0.001$ |
| R191A | No activity[a] |
| K196A | No activity[a] |
| E18A | No activity[a] |
| K75A | $0.19 \pm 0.01$ |
| R72A | $0.31 \pm 0.01$ |
| K68A | $0.69 \pm 0.04$ |
| K75A, R72A | $0.039 \pm 0.005$ |
| R72A, K68A | $0.15 \pm 0.07$ |
| K75A, R72A, K68A | $0.002 \pm 0.001$ |

Activity measurements were performed as described in Fig. 1. Turnover rates represent the average of 3 separate measurements ± s.d.
[a]No activity: turnover rate could not be calculated because no product could be detected or its level was below the detection limit of the assay

**Fig. 3** Binding of substrates at the catalytic site. **a** Interactions of PglH with the donor substrate UDP-GalNAc. Side chains of the C-terminal and N-terminal domains are colored pink or orange, respectively, and labeled in single-letter code. Black dashed lines indicate hydrogen bonds. **b** Stereo view of the catalytic site of PglH bound to LLO analog and UDP-CH$_2$-GalNAc. Acceptor and donor substrates are shown as blue and green sticks and labeled. PglH residues essential for function are shown as sticks and labeled. Black, dashed lines represent interactions between PglH and bound substrates. Red dashed lines indicate distances between the C4 hydroxyl of the acceptor GalNAc and the anomeric carbon or the methylene group from the phosphonate group. **c** Schematic of the interactions between PglH and donor and acceptor substrates in the catalytic site. A red dashed line indicates the distance between the C4 hydroxyl of the acceptor substrate and the anomeric carbon of the donor substrate. Black dashed lines represent interactions between PglH and the substrates. Green dashed lines represent two possible, but mutually exclusive hydrogen bonds

individually and in combination (Fig. 4b) by titrating the enzyme concentration. Individual mutations of K75 and R72 to alanine resulted in 10-fold and 6-fold reduction of the rate of hexa-LLO generation, respectively, and the combination of the two mutations led to a decrease in the turnover rate of about 50-fold (Table 1). Simultaneous mutation of R72 and K68 to alanine led to a 6.5-fold decrease in the turnover rate, whereas the simultaneous mutations of K75, R72, and K68 led to a 1000-fold decrease in the turnover rate of PglH. Quantification of the LLO intermediates of the reaction using single mutations showed that in the K75A mutant, an accumulation of the tri-LLO substrate (Und-PP-BacGalNAc$_2$) occurred (red bars in Suppl. Fig. 9, top left panel). In contrast, the mutant R72A yielded a slight accumulation of the tetra-LLO intermediate Und-PP-BacGalNAc$_3$ (green bars in Suppl. Fig. 9, top right panel). The strongest effect was observed for the mutant K68A, which resulted in a pronounced accumulation of the penta-LLO intermediate Und-PP-BacGalNAc$_4$ (yellow bars in Supplementary Fig. 9, bottom left panel). These findings are in line with a sequential interaction of the pyrophosphate group of the acceptor LLO with the residues K75, R72, and K68. When the activity measurements were performed with tri-LLO substrate reconstituted in liposomes, we found that the accumulation of intermediates was lower than in detergent, suggesting that the lipidic environment or the longer polyprenyl tail increase the processive characteristic of the reaction (Supplementary Fig. 10). For the PglH variant K75A, the tri-LLO substrate was accumulated as it was more slowly processed. Once the first GalNAc was added, the reaction of K75A proceeded quickly, as no other intermediates were observed. For R72A, no clear pattern was observed. For K68, a band corresponding to the penta-LLO intermediate was observed, albeit less pronounced than in the detergent-based reaction. In combination, our results suggest that the three positive charges in helix α2 are key elements of a molecular ruler that directly links the binding of the pyrophosphate group to successful binding of the LLO molecule for further addition of a GalNAc moiety. Upon addition of three GalNAc units, the pyrophosphate group would be pushed beyond the location of the three positively charged residues, resulting in decreased binding of the LLO substrate.

**Association of PglH with the membrane and LLO recruitment.** Given the amphipathic nature of the ruler helix (Fig. 2b) and its interaction with the pyrophosphate group of the LLO substrate,

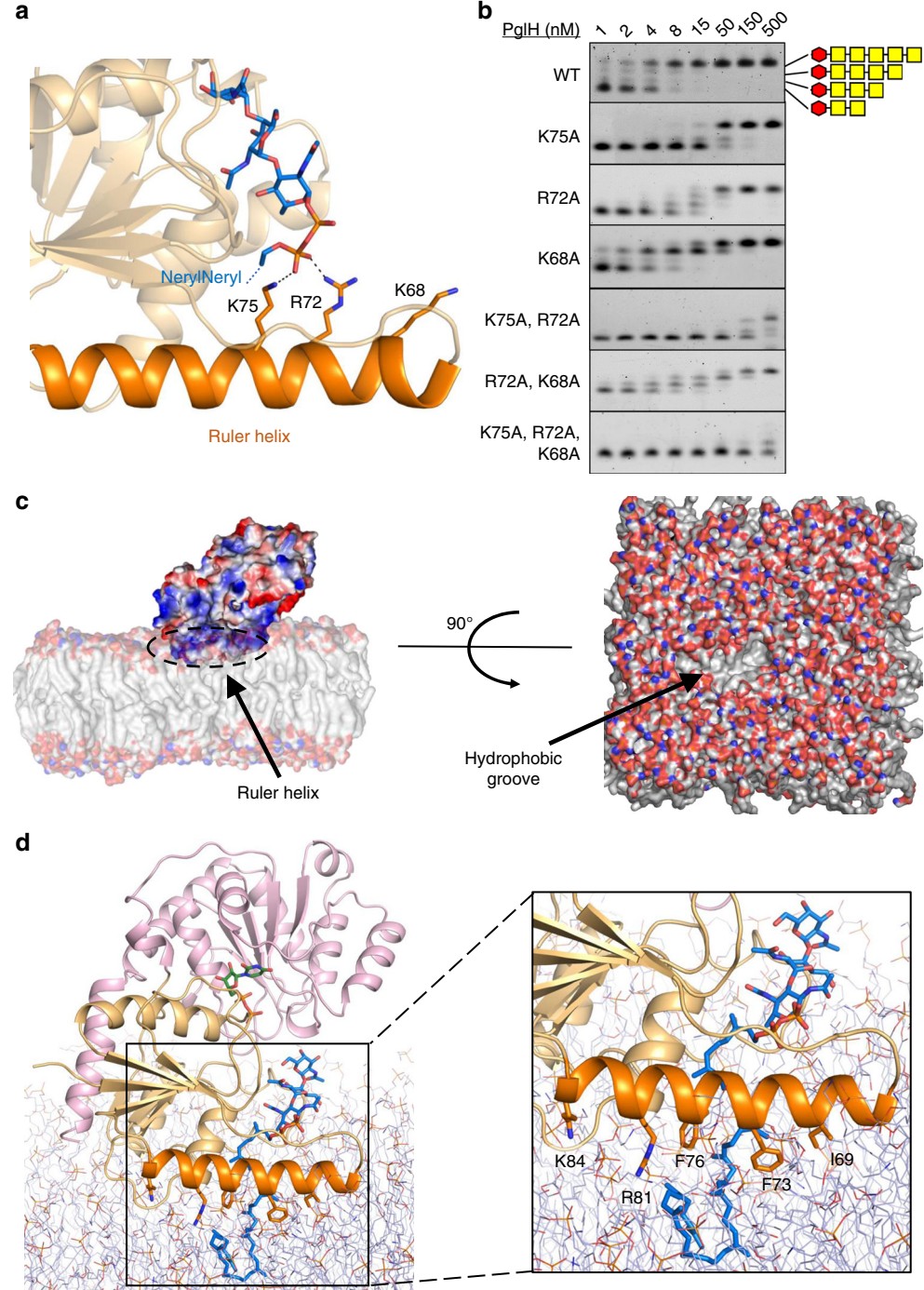

**Fig. 4** Dual role of ruler helix α2 in PglH. **a** Ribbon diagram of PglH bound to LLO analog as observed in the ternary complex PglH—UDP—LLO. The LLO analog and the three positively charged side chains of PglH interacting with the pyrophosphate group of the LLO are shown as sticks. Dashed lines depict salt bridge/hydrogen bond interactions. **b** Activity of wild type and mutant PglH. The reactions were performed for 10 min, after which the reactions stopped and the products were used as donor substrates for in vitro glycosylation of fluorescently labeled peptide by PglB and analyzed by tricine SDS-PAGE. The amount of enzyme varied and is indicated on top of the lanes. **c** Left, interaction of PglH with the modeled lipid bilayer. The electrostatic surface of PglH is shown, and the position of the ruler helix is indicated by and arrow and labeled. Lipids are shown in surface representation. Right, the hydrophobic groove generated in the membrane upon displacement of lipid head groups by PglH is shown by an arrow and labeled. **d** Interaction of PglH with the membrane bilayer and with the native LLO substrate Und-PP-Bac-GalNAc$_2$ as determined by molecular dynamics simulations. PglH is shown as a ribbon, with bound UDP and LLO shown as sticks. Phospholipids are shown as thin lines. Coordinates of the glycan and pyrophosphate moieties of LLO were from the ternary complex structure, whereas the polyprenyl tail was modeled. The inset shows a close-up of the interaction between the ruler helix with the lipid bilayer and LLO. PglH residues contributing to the interactions with lipid head groups or tails are shown as sticks and labeled

we hypothesized that it might interact with lipids and attach PglH to the membrane surface. To probe this interaction, we performed molecular dynamic simulations of PglH and a lipid membrane in the absence or presence of the polyprenyl tail of the native Und-PP-Bac-GalNAc$_2$ substrate. In the absence of LLO, PglH associated with the membrane in two similar orientations, both of which involve the ruler helix as the key element in membrane attachment, either by interaction of the positively charged residues R61, K74, R80, K81, K84, and K87 with lipid head groups (Supplementary Fig. 11a), or via hydrophobic interactions with the residues L65, I69, F73, and F77 (Supplementary Fig. 11b). Both orientations allow PglH to interact with the LLO substrate.

In simulations performed with the ternary complex with UDP and LLO, we found PglH to interact tightly with the lipid membrane. In particular, the hydrophobic face of the ruler helix was buried into the membrane and formed direct interactions with the lipid tails. The head groups of these lipids were pushed to the side, where they interacted with a series of basic residues forming a ring (Fig. 4c). During the simulation, the glycan moiety of the LLO was not as close to the active site as observed in the crystal structures. However, three isoprenyl units of the LLO interacted with F76 and F73 of the ruler helix (Fig. 4d). A combination of hydrophobic and charge interactions of lipids with the ruler helix and additional side chains of the N-terminal domain thus stabilize PglH at the membrane, with its active site accessible to membrane-anchored LLO.

## Discussion

The structural characterization of membrane-associated GTs in complex with substrates is particularly challenging if the substrates are non-commercial and contain long lipidic tails that limit water solubility. The chemo-enzymatic generation of tri-LLO analogs combined with the synthesis of the non-hydrolyzable phosphonate analog of UDP-GalNAc allowed for the first time the trapping and visualization of a transition-state mimic of a membrane-associated GT-B. Similar analogs of nucleotide-activated sugars had previously been used as inhibitors of GTs or have been co-crystallized with GTs that exhibit glycosidase activity[29–32]. However, ternary complexes containing acceptor and donor substrate analogs have been only reported for retaining GT-B members with soluble substrates such as MshA (in complex with UDP and 1-L-myo-inositol-1-phosphate bound)[26], OtsA (in complex with Glc-6-P and UDP)[33], and GtfA (in complex with vancomycin and nucleotide diphosphate bound)[34].

Retaining GTs have been proposed to either follow a double displacement mechanism, involving the formation of an intermediate covalently bound to the enzyme, or a SNi-like mechanism, which requires the formation of an oxocarbenium intermediate[20, 28, 35, 36]. Similar to observations in the GT-B members BshA[37], MshA[26], and OtsA[24, 33], our PglH structure did not reveal a nucleophile in the active site that would facilitate a double displacement mechanism. Instead, PglH might indeed follow a $S_N1$-like mechanism, where the nucleophilic attack from the acceptor substrate and the departure of the leaving UDP occur from the same side. This would require the leaving group to act as a catalytic base to activate the incoming nucleophile[38].

Based on the modeling of a ternary complex followed by quantum mechanics calculations, a mechanism for glucose transfer from UDP-Glc to glucose-6-phosphate by the retaining GT OstA was recently proposed[39]. In this study, a hydrogen bond between a hydroxyl group of the acceptor substrate and one of the non-bridging oxygen atoms of the pyrophosphate moiety of UDP was postulated. In our PglH ternary complexes, we indeed observed

such a hydrogen bond between the C4 hydroxyl of the acceptor GalNAc and UDP (Fig. 3b, c). For OstA, a second hydrogen bond between the acceptor hydroxyl and the oxygen connecting the anomeric carbon with the first phosphate group was proposed to stabilize the leaving UDP and subsequently enhance the nucleophilicity of the acceptor hydroxyl. Because we used a phosphonate analog (Fig. 3b, c), we do not have structural evidence of a similar hydrogen bond having a role in PglH. However, the observed distance would be compatible with the formation of a hydrogen bond if an oxygen was present instead of the methylene group. The last step in the proposed OstA mechanism is the nucleophilic attack from the acceptor hydroxyl to the $sp^2$ anomeric carbon of a short-lived oxocarbenium intermediate. We observed the C4 hydroxyl of the acceptor GalNAc positioned at ~2.7 Å from the anomeric carbon of UDP-CH$_2$-GalNAc (Fig. 3b, c). Taking in account the hydrogen bound to the $sp^3$ anomeric carbon, this distance would be lower than the combined van der Waals radii of the two atoms. However, given the estimated coordinate error for this structure (maximun-likelihood based, 0.41 Å), the real distance might be longer. Nevertheless, our structure probably represents a state prior to the formation, but structurally similar to the transition state of the reaction.

In various biological processes, the length of lipid-linked glycan polymers is controlled by elongation termination. In the biosynthesis of the *E. coli* O9a antigen, glycan polymerization is catalyzed by the mannosyltransferase WbdA, and the length of the growing glycan chain is controlled by the C-terminal domain of the associated WbdD protein, which adopts a coiled-coil folding and acts as a molecular ruler[40]. In the biosynthesis of bacterial lipopolysaccharide, glycan repeats are polymerized by the integral membrane protein Wzy, and the length of the polymer is regulated by Wzz, using an unknown mechanism[41]. In contrast, PglH is not an integral transmembrane protein and does not require an additional partner to perform the glycan counting. Instead, it contains a ruler helix that also mediates its association to the plasma membrane. This helix and its interaction with membrane lipids has been proposed for the related GTs GumK[18], MGD1[19], and MurG[42]. For PimA, it was shown that mutations in a cluster of positive residues interacting with head groups of the lipid bilayer reduced the activity in vivo, despite the fact that in vitro binding of donor substrate was unaffected[17, 25]. For successful LLO binding and GalNAc transfer, a salt bridge/hydrogen bond interaction (3 − 4 kcal mol$^{-1}$ per interaction) between the pyrophosphate group of the LLO and the ruler helix residues is required, and at the same time access of the terminal GalNAc to the catalytic site of PglH. Due to the rigid, rod-shape glycan structure[43], only a penta-saccharide LLO substrate can be fit into the active site, preventing efficient elongation in vivo beyond the addition of three GalNAcs. The counting mechanism we propose for PglH therefore depends on the interaction between the ruler helix and the LLO substrate and the rigidity of the glycan structure. Once released, hexa-saccharide LLO is further modified by PglI, adding a branching glucose moiety (Fig. 5). Once this is added, LLO binding to PglH is impossible due to steric clashes. The counting mechanism of the processive, membrane-associated GT PglH might be shared by other GT-B GTs, given that there is a high structural conservation among members of this protein family.

## Methods

**PglH expression and purification**. A synthetic gene encoding *C. jejuni* PglH (Supplementary Methods) was cloned into a modified pET-19b vector (Novagen) with a His$_{10}$ affinity tag fused to the C terminus. PglH mutants were generated by using gBlocks gene fragments (Integrated DNA Technologies) and the resulting plasmids of all constructs were sequenced (Microsynth). PglH wild type and mutants were overexpressed in *E. coli* BL21-Gold (DE3) (Stratagene). Cells were grown at 37 °C in modified Terrific Broth (TB) medium supplemented with 1%

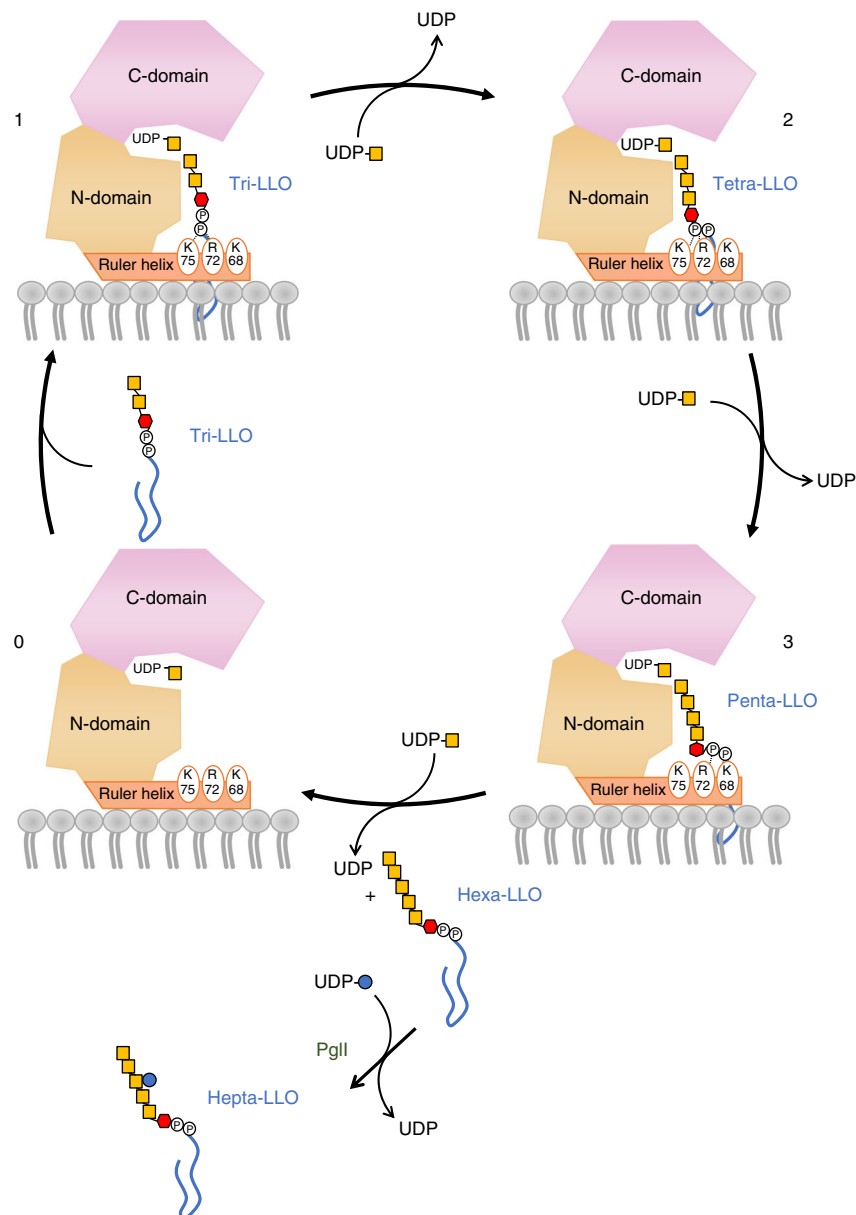

**Fig. 5** Glycan counting mechanism during PglH reaction. PglH is shown schematically, with N-terminal and C-terminal domains shaded yellow and pink, respectively, and labeled. The ruler helix is shaded orange, and the three residues relevant to the counting mechanism are indicated. Phospholipids from leaflet of the lipid bilayer are depicted in gray. The polyprenyl tail of the LLO is depicted as a blue, curved line, phosphate moieties as a circled P, and the saccharide symbols are as in Fig. 1. State 0 corresponds to UDP-GalNAc-bound PglH as observed in our crystal structure. State 1 represents a ternary complex upon binding of Und-PP-Bac-GalNAc$_2$ (tri-LLO), with the pyrophosphate group of tri-LLO interacting with the side chains of residues K75 and R72 as observed in the two crystal structures with LLO bound. Following transfer of the first GalNAc, the product Und-PP-Bac-GalNAc$_3$ (Tetra-LLO) is predicted to move along the ruler helix, where the pyrophosphate moiety of the LLO will be coordinated mainly by the side chains of R72 (state 2). After the second transfer reaction, the product, Und-PP-Bac-GalNAc$_4$ (Penta-LLO) is predicted to move along the ruler helix once more, where the pyrophosphate moiety of the LLO will be coordinated mainly by K68 (state 3). The shown interactions of the LLO with the ruler helix are inferred from functional experiments. Each transfer reaction involves an exchange of UDP for a UDP-GalNAc. Once the product Und-PP-Bac-GalNAc$_5$ (Hexa-LLO) is released from PglH, it is processed by PglI, which adds a glucose moiety to the third GalNAc

glycerol (w/v) to A600 nm of 2.7 before expression was induced by the addition of 1 mM isopropyl β-D-1-thiogalactopyranoside (IPTG) for 16 h at 18 °C. All following steps were performed at 4 °C. Cells were collected by centrifugation and resuspended in 50 mM Tris-HCl, pH 8.0; 200 mM NaCl; 3 mM β-mercaptoethanol; 3% glycerol; and 0.5 mM phenylmethanesulfonylfluoride (PMSF). Cell lysis was performed in a M-110L microfluidizer Microfluidics) at 15,000 p.s.i. chamber pressure. Membranes were pelleted by ultracentrifugation at 100,000×g for 30 min.

PglH was extracted from membranes in 50 mM Tris-HCl, pH 8.0; 200 mM NaCl; 25 mM imidazole; 3% glycerol (v/v); 3 mM β-mercaptoethanol; 1% Triton X-100 (w/v) by stirring for 1 h. The supernatant was loaded onto a NiNTA superflow affinity column (Qiagen), followed by washing with 25 column volumes with the

same buffer but containing 0.02% N-dodecyl-β-D-maltopyranoside (w/v) (Anatrace) and 50 mM imidazole. Elution was performed in the same buffer but containing 200 mM imidazole. The protein was desalted into 10 mM Tris-HCl, pH 8.0; 150 mM NaCl; 3% glycerol; 0.5% Polyethylene glycol 1000 (w/v); and 2 mM dithiothreitol (DTT) and concentrated to a final volume of 500 μL in an Amicon Ultra-15 concentrator (Millipore) with a molecular mass cutoff of 50 kDa. The protein was further purified by size exclusion chromatography (Superdex 200 10/300 GL, GE Healthcare) and the buffer was exchanged to 10 mM Hepes, pH 7.5; 150 mM NaCl; 3% glycerol; 0.5% Polyethylene glycol 1000 (w/v); and 2 mM DTT. Peak fractions were pooled and concentrated to 10–12 mg mL$^{-1}$ in an Amicon Ultra-15 concentrator (Millipore) with a molecular mass cutoff of 50 kDa.

**Selenomethionine derivative production**. Production of selenomethionine derivative was performed as described by Perez et al.[8]. In brief, *E. coli* BL21 (DE3) (Stratagene) expressing PglH were grown in modified TB medium supplemented with 1% glycerol (w/v), at 37 °C to A600 nm of 0.5–1.0 and used to inoculate a culture of M9 media supplemented with vitamin B1 hydrochloride. Cells were then grown until A600 nm of 0.5 and used to inoculate a new culture of M9 media supplemented with vitamin B1 hydrochloride. Cells were grown overnight at 37 °C until A600 nm of 0.9. At that point a cocktail of amino acids (lysine, threonine, phenylalanine, leucine, isoleucine and valine, all at 100 mg L$^{-1}$) including selenomethionine (200 mg L$^{-1}$) was added to the culture. After 30 min, expression was induced with 1 mM IPTG for 90 min. Cells were collected by centrifugation and preparation of membrane fraction was performed as described above.

**Crystallization of PglH**. For crystallization of PglH (native or the selenomethionine derivative) with bound donor substrate, concentrated PglH was incubated with UDP-GalNAc to a final concentration of 2 mM for 15 min. Crystallization was performed by vapor diffusion in sitting or hanging drops at 20 °C, using a reservoir containing 0.2 M KI; 0.1 M Mops 7.2; and 17% PEG3350. The volume ratio of protein to reservoir was 1:1.

For crystallization of PglH with bound acceptor substrate, concentrated PglH was incubated with synthetic LLO at a final concentration of 0.6 mM and UDP or UDP-CH$_2$-GalNAc at a final concentration of 2 mM for 15 min. Crystallization was performed by vapor diffusion in sitting drops or hanging drops at 20 °C using a reservoir containing 0.2 M Ammonium acetate; 0.1 M ADA 6.3; and 21–22% PEG3350. The volume ratio of protein to reservoir was 1:1.

Crystals typically appeared after 1 day and matured to full size within 3–4 days. Crystals were cryoprotected by gently increasing the cryoprotectant concentration in the drops (up to 30% PEG3350) and directly flash frozen by immersion in liquid nitrogen before data collection.

**Data collection and structure refinement**. Crystals of all three structures belonged to the space group P2$_1$. Native and selenomethionine derivative data were collected at the beamline X06SA at the Swiss Light Source (SLS, Villigen). For PglH bound to UDP-GalNAc, phases were solved by running Crank2[44] with SAD data. The overall figure of merit was 0.82 and eight selenium sites were found in the asymmetric unit. The initial model was built using Coot[45]. The structures of PglH in complex with LLO acceptor substrate were determined by molecular replacement with PglH bound to UDP-GalNAc. Two copies of PglH were found in the asymmetric unit and noncrystallographic symmetry restraints were initially used for the refinement. For the highest-resolution structure (UDP-GalNAc), these restraints were released in the later stages of the refinement. For the lower-resolution LLO-complexes, NCS restraints were maintained. Iterative model building was performed in Coot[45] and the X-ray structure was refined in PHENIX[46]. The final structures of PglH/UDP-GalNAc, PglH/UDP/LLO, and PglH/UDP-CH$_2$-GalNAc/LLO have 96,94%, 96.28%, and 88.13% of Ramachadran favored; 2.65%, 2.79%, and 10.20% of Ramachadran allowed; and 0.42%, 0.98%, and 1.68% of Ramachandran outliers, respectively.

**Cloning and expression and purification of PglA and PglJ**. The genes encoding PglA and PglJ from *C. jejuni* (Supplementary Table 2) were cloned into a modified pET-19b vector (Novagen) with a His$_{10}$ affinity tag fused to the C terminus. PglA and PglJ were overexpressed in *E. coli* BL21-Gold (DE3) (Stratagene). Cells were grown at 37 °C in modified TB medium supplemented with 1% glycerol (w/v) to A600 nm of 2.7 before expression was induced by the addition of 1 mM isopropyl IPTG for 3 h at 37 °C. All following steps were performed at 4 °C. Cells were collected by centrifugation, resuspended in 50 mM Tris-HCl pH 8.0; 200 mM NaCl; 3 mM β-mercaptoethanol; 3% glycerol; and 0.5 mM PMSF. Cell lysis was performed in a M-110L microfluidizer Microfluidics) at 15,000 p.s.i. chamber pressure and the cell lysate was incubated with a final concentration of 0.5% Triton X-100 for 1 h. Cell debris and insoluble material was pelleted by ultra-centrifugation at 100,000×*g*, in Ti45i rotor, for 30 min. The supernatant was supplemented with 25 mM imidazole and loaded onto a NiNTA superflow affinity column (Qiagen, Hilden, Germany), washed extensively with the same lysis buffer but containing 50 mM imidazole and 0.05% Triton X-100. Elution was performed in the same buffer but containing 200 mM imidazole. The proteins were desalted into 25 mM Hepes pH 7.5; 150 mM NaCl; 5% glycerol, 2 mM DTT; and 0.05% Triton X-100 using a HiPrep 26/10 column (GE Healthcare, Little Chalfont, UK).

**Chemo-enzymatic preparation of synthetic LLO substrate**. The chemical synthesis of the LLO analog NerylNeryl-PP-GlcNAc included three steps: the synthesis of GlcNAc α-phosphate, the preparation of the lipid phosphate precursors, and the coupling of both monophosphates to the desired final product[16]. The enzymatic addition of GalNAc units was performed in the same desalting buffer for PglA and PglJ, but supplemented with 2 mM MgCl$_2$. The first reaction catalyzed by PglA was performed incubating mM NerylNeryl-PP-GlcNAc with UDP-GalNAc in a 4:1 molar ratio (UDP-GalNAc:LLO) and purified PglA. The reaction product was purified by freeze drying of the reaction mixture followed by CHCl$_3$/MeOH (2:1, v/v) extraction. The organic phase was dried in a nitrogen stream and the LLO products resuspended in the reaction buffer for the next

reaction. The second addition of GalNAc and the purification of the final product was performed in the same way. The concentration of the final synthetic LLO substrate was determined by titrating different amounts of LLO against a constant amount of the fluorescently labeled acceptor peptide 5CF- GSDQNATF in an in vitro glycosylation assay with purified PglB[14].

**Chemical synthesis of UDP-CH$_2$-GalNAc**. The preparation of UDP-CH$_2$-GalNAc was designed based on the synthesis of the C4-epimer UDP-CH$_2$-GlcNAc[30]. Several reaction conditions were optimized to achieve higher yields and purities such as removal of the oxazoline side product by acidic treatment after the radical allylation step, diesterification of the hydroxy phosphonate using pyridine and 4-dimethylaminopyridine (DMAP) as base, quick dealkylation with an excess of TMSBr, and deacetylation with ammonium hydroxide. Finally, purification of the final product was performed by size exclusion chromatography eluted with 0.25 M NH$_4$HCO$_3$ buffer[47], yielding UDP-CH$_2$-GalNAc in 40 mg scale and in pure form as confirmed by nuclear magnetic resonance spectroscopy and high-resolution mass spectrometry. Detailed reaction conditions and spectroscopic characterization can be found in Supplementary Methods.

**LLO extraction**. Isolation of LLO was performed as described previously[8]. In brief, LLO was extracted from *E. coli* SCM6 cells carrying a *C. jejuni pgl* cluster, containing inactivated *pglB* and *pglH* genes. Lipid extraction was performed using a mixture of methanol/chloroform (1:2), dried in a rotary evaporator, and reconstituted in a buffer containing 10 mM Tris, pH 8.0, 150 mM NaCl, and 1%Triton X-100 (w/v). The concentration of reconstituted LLOs was determined as described before.

**In vitro activity assays for PglH**. Reaction mixtures usually contained 1–500 nM purified PglH protein, 20 µM LLO substrate (native or synthetic), 100 µM UDP-GalNAc, 150 mM NaCl, 20 mM Hepes pH 7.5, 2 mM DTT, and 1% Triton X-100. Reactions were incubated at 25 °C for 10 min and stopped by incubation for 1 min at 75 °C. Reaction products were used as donor substrates for transfer onto a fluorescently labeled peptide using purified PglB, followed by tricine SDS-PAGE analysis[48]. Fluorescent bands for peptide and glycopeptides were visualized by using a Typhoon Trio Plus imager (GE Healthcare) with excitation set at 488 nm and using a 526 nm SP emission filter. The amount of formed glycopeptide was determined from band intensities of fluorescence gel scans (ImageJ)[14]. Uncropped scans from the gels show in Figs. 1 and 4 are shown in Supplementary Figs. 12 and 13.

For activity measurements in liposomes, these were generated using a 3:1 (w:w) mixture of *E. coli* polar lipids and L-α-phosphatidilcholine (AvantiPolar Lipids) and native tri-LLO was reconstituted[8]. Briefly, liposomes were prepared by extrusion through polycarbonate filters (400 nm pore size) and diluted in 10 mM Hepes, pH 7.5, 150 mM NaCl, and 2 mM DTT. After saturation with Triton X-100, the liposomes were mixed with native Tri-LLO. BioBeads were then added to remove detergent. Finally, the concentration of reconstituted Tri-LLO was determined by titrating various amounts of liposomes against a constant amount of acceptor peptide in an in vitro glycosylation assay using purified PglB. Reaction mixtures contained 50, 250, or 500 nM purified (wild type or mutant) PglH protein, 1 µM LLO substrate, 10 µM UDP-GalNAc, 150 mM NaCl, 20 mM Hepes pH 7.5, and 2 mM DTT. Reactions were incubated and the products detected as described above for detergent-based activity assays.

For inhibition analysis, reaction mixtures containing 2 nM PglH protein, 20 µM Und-PP-Bac-GalNAc$_2$, 150 mM NaCl, 20 mM Hepes pH 7.5, 2 mM DTT, and 1% Triton X-100 were incubated with different concentrations of UDP or UDP-C-GalNAc (15 nM to 8 mM) for 5 min at 25 °C. Reactions were started with addition of 100 µM of UDP-GalNAc. Samples were taken after 10 min. Data were fitted to non-linear regression and IC$_{50}$ was determined using PRISM software.

**Molecular dynamics simulations**. The interaction of PglH with a lipid membrane was studied with all-atom explicit solvent molecular dynamics (MD) simulation using GROMACS 5.0.6.[49] Initially, a bilayer of 392 palmitoyl oleoyl phosphatidyl-ethanolamine (POPE) and 98 palmitoyl oleoyl phosphatidyl-glycerol (POPG) lipids was created using the CHARMM-GUI webserver[50]. This system was solvated with water and 150 mM NaCl, resulting in a box of ~12 × 12 × 17 nm$^3$. After 50,000 steps of steepest descent energy minimization, the membrane patch was simulated for 259 ns of MD simulation in the isothermal-isobaric (NPT) ensemble. Briefly, the initial systems were equilibrated for 11 ns of MD simulation in the NPT ensemble in which all non-hydrogen atoms of the protein were restrained to the fixed reference positions with progressively smaller force constants, starting at 1000 kJ mol$^{-1}$ nm$^2$. Periodic boundary conditions were used. Particle mesh Ewald[51] with cubic interpolation and 0.16 nm grid spacing for Fast Fourier Transform was used to treat long-range electrostatic interactions. The time step was 2 fs. The LINCS algorithm[52] was used to fix all bond lengths. Constant temperature (310 K) was set with a Nose–Hoover thermostat[53], with a coupling constant of 0.5 ps. A semiisotropic Parrinello–Rahman barostat[54] was used to maintain a pressure of 1 bar

The final structure of the lipid membrane simulation was used as starting point for PglH-membrane docking simulations. PglH bound to UDP-GalNAc (PDB ID

6EJI) was placed 1 nm away from the membrane. Ten random PglH orientations were created using rigid-body rotation. Each of these configurations was initially equilibrated for 10 ns. Restraints on non-hydrogen atoms of the protein were gradually released, starting from 1000 kJ mol$^{-1}$ nm$^2$. Then, 90 ns of MD was performed for each setup. Four of the runs, in which PglH had approached the membrane, were extended to 150 ns. Of these, the two runs showing protein-membrane interactions were extended further for a total of 250 ns each. In one simulation, PglH docked tightly into the membrane surface.

For the simulation of PglH bound to UDP and tri-LLO, the structure bound to the synthetic LLO substrate was used (PDB ID 6EJJ). The synthetic LLO was replaced by the native Und-PP-Bac-GalNAc$_2$. The whole complex was aligned with the end structure from the simulation of PglH with UDP-GalNAc bound and was placed in the same position relative to the membrane. The tail of LLO was built and placed in the membrane. After minimization and 10 ns of equilibration, 500 ns of unconstrained simulation were performed.

The all-atom CHARMM36 force field was used for protein, lipids, and ions, and TIP3P was used for water molecules[55, 56]. The MD trajectories were analyzed with Visual Molecular Dynamics (VMD)[57].

**Data availability**. Atomic coordinates and structure factors were deposited in the RCSB Protein Data Bank (PDB) under accessions 6EJI (PglH in complex with UDP-GalNAc), 6EJJ (PglH in complex with UDP and synthetic LLO), and 6EJK (PglH in complex with UDP-CH$_2$-GalNAc and synthetic LLO). The data that support the findings of this study are available from the corresponding author upon reasonable request.

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

## Acknowledgements

We thank the staff scientists at the PX beamline of the Swiss Light Source for help with data collection, Maja Napiórkowska for providing purified PglB for in vitro activity assays, Daniel Janser for help with protein expression, the Mass Spectrometry Service (Department of Chemistry and Biochemistry, University of Berne) for recording MS spectra of synthetic LLO substrates, Max Linke for providing the code for the rigid-body rotation process, and Markus Aebi for helpful discussions. This work was supported by the Swiss National Science Foundation (Transglyco Sinergia program to J.-L.R. and K.P.L.) and ETH Zurich grant 27-16-2 to K.P.L.. A.R.M and G.H. acknowledge support from the German Science Foundation (SFB 807, CEF Macromolecular Complexes) and from the Max Planck Society.

## Author contributions

A.S.R. performed cloning, expression, purification of PglH wild type and mutants, determined the structures of PglH, and performed the in vitro activity assays. J.B. performed chemical synthesis of substrate analogs. A.R.M. performed molecular dynamic simulations. G.H. supervised molecular dynamic simulations. T.D. and J.-L.R. supervised the chemical synthesis of substrate analogs. K.P.L. and A.S.R. conceived the project and wrote the manuscript.

## Additional information

**Competing interests:** The authors declare no competing financial interests.

