## [Peer Review File · Nature Communications]

Editorial Note: This manuscript has been previously reviewed at another journal that is not operating a transparent peer review scheme. This document only contains reviewer comments and rebuttal letters for versions considered at Nature Communications. Mentions of prior referee reports have been redacted.

Reviewers' comments:

Reviewer #1 (Remarks to the Author):

Ramírez and co-workers report crystal structures of the processive membrane associated glycosyltransferase PglH from *Campylobacter jejuni*, an enzyme that catalyzes the transfer of three GalNAc residues to the growing lipid-linked oligosaccharide (LLO) precursor, undecaprenyl-PP-Bac-GalNAc₂. Specifically, the authors present three crystal structures of PglH: a binary complex with the sugar donor UDP-GalNAc at 2.3Å, and two ternary complexes with a chemo-enzymatically generated LLO analog and either UDP or nonhydrolyzable UDP-CH₂-GalNAc, at 2.7 and 3.3Å, respectively. It is worth noting that the two ternary complexes represent the first crystal structures of a GT-B enzyme in the presence of lipid acceptor substrate/derivative. PglH displays a typical GT-B fold, consisting of two Rossmann-fold domains, with an important fissure in which UDP-GalNAc and the LLO analog are located. PglH displays a long amphipathic alpha helix in the N-terminal domain, which mediates membrane association. Interestingly, this alpha helix displays three positively charged residues that interact with the pyrophosphate group of the LLO acceptor substrate analog. In combination with enzymatic and molecular dynamics studies the authors propose a molecular ruler mechanism for glycan counting. Finally, the PglH/UDP-CH₂-GalNAc/LLO analog ternary complex suggest/support that PglH uses an SN₁-like mechanism. The article is clear and well written. Because of its novelty and the quality of the work, I strongly support the publication of this manuscript in Nature Communications.

Some questions to be addressed and suggestions for improvement are listed below:

1. page 2: "Here, we present crystal structures of PglH in three distinct states". A good opportunity to describe that PglH is a retaining glycosyltransferase.
2. page 2: "PglH catalyzes the transfer of exactly three N-acetylglalactosamine (GalNAc) units to the growing LLO precursor". Please, describe the sugar linkages between GalNAc residues transferred by PglH.
3. page 3: "Whereas PglA and PglJ each catalyzes a single GalNAc transfer reaction⁶, PglH catalyzes the addition of three GalNAc units". Please, describe the sugar linkages between GalNAc residues transferred by PglA, PglJ and PglH in the Introduction. Please add this information in Figure 1a and Supplementary Figure 1.
4. page 4: "Although *C. jejuni* PglH is not an integral membrane protein, non-ionic detergents (Triton X-100 and dodecyl maltoside) were required for its successful purification in homogeneous and active form". What is the criteria to classify PglH as a non-integral membrane protein? Please add references accordingly. Any trace of protein in the soluble fraction?
5. page 18: Figure 2a. Please add a panel where the PglH-UDP-C-GalNAc is rotated 90 degrees – with the ruler helix perpendicular to the current view.
6. Supplementary Information:
 - The complete amino acid sequence of the PglH construct should be included as a Supplementary Figure 2. In the case the authors used a synthetic pglH gene, with modified/improved DNA

sequence, this information should be also included in Supplementary Figure 2.

- Please include experimental phasing data in Table S1.

7. Spelling:

- "three N-acetylglalactosamine" by "three N-acetylgalactosamine",

- "with the acceptor substrate UDP-GalNac" by "with the donor substrate UDP-GalNac",

Reviewer #2 (Remarks to the Author):

The issue of polymerization control is of high interest in glycobiology. Nature makes polymers of defined length that would be hard to achieve by simple chemical means. The paper discusses a system for controlling lipid-linked oligosaccharide polymerization by limiting to three the number of UDP-GalNac added to a trisaccharide. The paper presents the following experimental data,

- 1 The structure of the enzyme bound to UDP, UDP-CH₂-GalNac with LLO mimic (short lipid but trisaccharide)

- 2 The mutagenesis of a number of residues on the key helix (ruler).

- 3 Detailed modeling studies.

The structures reveal the active site and show the pyrophosphate of the LLO binds to R72 and K75.

Mutagenesis shows that K75A, R72A slow the enzyme. The double mutant K75A, R72A is very slow.

The most pertinent residue to their model for the ruler is K68. Mutation of K68 results in accumulation of pentasaccharide.

The thesis which is strongly supported by modeling is that the phosphate group jumps along the positively charged residues. In doing so it allows the substrate to grow by one residue at a time until the limit. In this model, the enzyme binds the tri, tetra and penta sugars (Figure 5).

This is interesting and novel. Further experimental support would strengthen the paper. The K68A mutant does not appear to affect 4 to 5 step, yet it is proposed to be important in Figure 5. Why not? What does the K68R look like? It is also puzzling that since the K68 PP is proposed to be such a strong interaction (prevent competition of intermediates) why when it is abolished does the enzyme not go onto to create a 7 mer? If it creates the 6 mer without an anchor, why not longer?

In the model, the reducing sugar is pushed into a pocket in the tetra LLO. Have the authors considered trying to block the motion of the reducing sugar or otherwise block the re-arrangement of the LLO? Being able to identify such a mutant which would stall polymerization.

This is a good piece of work that deserves to be published in Nat Comm. If Nat Comm is to remain of the highest impact then given it's not the first ruler to be described (the Rock kinases should be cited) then pushing for that nit extra experimental validation is important. With that additional data, this paper will be seen as a landmark in disclosing a new mechanism of polymerization control.

Reviewer #3 (Remarks to the Author):

Ramirez et al report the synthesis of a soluble substrate analog and development of an enzymatic assay to measure the turnover of PglH glycosyltransferase *in vitro*. They determined the structure of PglH, showing has an unusually long helix 2, with exposed basic residues. They determined this structure in the presence of the donor, as a product analog ternary complex, and as a reaction ready ternary complex inhibited by the presence of a non-reactive donor analog. Molecular dynamics simulations was used to explore the mode of interaction of PglH with the membrane, and to study possible interactions of the substrate with PglH. Finally, the authors mutate the three basic residues in helix 2 individually and in combination and show that these affect the relative accumulation rates of different products.

Overall the paper represents significant progress in our understanding of this enzyme in particular, and of the strategies that bacterial polysaccharide glycosyltransferases can employ to build specific glycostructures. A broad range of data are presented to support the paper's central ideas, with the approaches appropriate and the data appearing technically sound. The explication of the ternary complexes is a notable technical achievement, and the molecular ruler idea interesting, novel, and well presented. The methods are well enough documented to allow their replication by others in the field. The paper is well written, and argues its points clearly and succinctly. I believe that the implications are broad enough to interest the general readership of *Nature Communications*. My main quibble is that there is, in my view, some nuance missing around their data analysis and model of the molecular ruler hypothesis.

1) The reaction is carried out in the solution phase with a relatively soluble version of the substrate. While I fully appreciate the technical necessity, this alteration could reasonably result in behavior in this assay that does not reflect the behavior in cell. For example, the kinetics in solution reflect a three-dimensional search between acceptor and substrate, while the search in the membrane would be approximately two dimensional. In addition, in a lipid bilayer, phospholipids would compete with the substrate for charge-charge interactions with the basic side chains. And in solution, with no competing lipid tail interactions, the neryl-neryl group may interact with exposed hydrophobic patches on the protein, resulting in increased affinity and a possibly strong localization effect. I acknowledge that the ability of the solution assay to recapitulate the main outputs of the *in vivo* assay is a strong point in its favour, and the effect of the basic residue point mutations is strong, but I would still be more comfortable with this result if the authors were able to recapitulate the essential finding *in vivo* – e.g. by showing that the triple basic mutant fails to rescue a PglH knockout. At the very least, the limitations of interpreting the subtleties of a solution assay that stands in for a membrane embedded system should be clearly acknowledged in the discussion.

2) Given that the mutation of the three basic residues is argued to affect the turnover of each of the three distinct substrates separately, ideally the data shown in fig 4b should be analyzed in terms of reaction rates with each distinct substrate (rather than aggregated into a single turnover number). Plotting the fraction of each substrate that has proceeded through a given step in the reaction as a function of enzyme concentration might allow the maximal rate of each step to be at least estimated for each construct. Quantifying the effect of each mutation on each step would allow a little more nuance in terms of discussing the mechanism of chain length control. In particular, I suspect that all three residues have a measurable effect on all three steps of the reaction (e.g. by additional interactions with the GalNac residues by R72/K75 in longer substrates). An alternative test of the idea that the sites are independent would be to test whether an equimolar mix of K75 protein plus K68 protein can act as effectively as the wild type.

3) The presentation of the mechanism of the molecular ruler is perhaps somewhat simplistic. For example, the basic side chains should have a fairly large degree of flexibility that would allow them to reorganize to accommodate different residue positions. Possibly a bidentate interaction with K68/Arg72 may still be possible, even with the GalNac4 substrate. The text also in some places implies a one to one correspondence between the basic residues on helix 2 and the GalNac

residues added. E.g. Abstract "The ruler helix contains three positively charged side chains that can bind the pyrophosphate group of the LLO substrate and thus limit the addition of GalNAc units to three."

4) The PglH/UDP-CH₂-GalNAc/LLO structure has very high average ADPs for the ligand in comparison to the rest of the structure. While the authors make clear that not all of this ligand is clear in the electron density map, it might be useful to discuss the limitations of interpreting this map in a little more detail. In particular, the pyrophosphate density from this map is the only direct evidence that the critical basic residues are required to bind these phosphate groups, so the quality of this evidence is a critical building block of the mechanism proposed.

Additional points:

It is not accurate to refer to the PglH/LLO/UDP complex as a "product complex" (e.g. in the results). The LLO analog used has only 2 GalNAc residues added, and therefore cannot result from a PglH reaction. In addition the location of the LLO seems very similar to the positioning as required to act as an acceptor, rather than with the terminal GalNAc near where the GalNAc portion of UDP-GalNAc sits in the donor complex; a substrate with a single GalNAc (whose reaction would result in this product) would presumably not be able to reach from the basic helix to the active site.

Pg 7. "and ~3.4 Å from the methylene group" – I had to think for 5 seconds to figure out which methylene was being referred to. Maybe be more explicit – e.g. "the methylene group mimicking O4".

Pg 7 – Arg191 is argued to stabilize the leaving group on pg 5, with this idea being supported using the same the strong phenotype of a point mutation. While these experiments do not formally distinguish which function is the more critical in abrogating function, the wide conservation of this residue would argue that its role may not be primarily in binding the acceptor glycosyl group. I think it would be fairer to state that the importance of the Arg191 C2 acetamido interaction is not easily addressable by mutagenesis given its multiple functional roles.

Fig 4a – In the figure caption, maybe indicate that this is the experimental structure, not an MD frame.

Fig 5 caption: Wording of the mechanism implies that the reaction is precessive, with substrate being transferred between sites rather than being released between steps. No data is presented to prefer such a mechanism, and the accumulation of intermediates in even the wild type enzyme would seem to argue against it.

Table 1 – it would be more useful to estimate the lower bound of activity detectable by the assay than state that no product was detected.

Methods - Pg 26. Is the described amino acid mix the stock concentration, or the final concentration. These values seem unreasonably high (total of 800 g / L total amino acids)

I understand that refining structures at low resolution is somewhat challenging, but the Ramachandran plot values for the PglH/UDP-CH₂-GalNAc/LLO ternary complex seem a little poor, given that they are based on a well refined higher resolution binary structure.

We thank the reviewers for their insightful and constructive comments. In the following, reviewer comments are in italics and our responses / actions are in plain text colored red.

1. Reviewer #1:

Ramírez and co-workers report crystal structures of the processive membrane associated glycosyltransferase PglH from Campylobacter jejuni, an enzyme that catalyzes the transfer of three GalNAc residues to the growing lipid-linked oligosaccharide (LLO) precursor, undecaprenyl-PP-Bac-GalNAc₂. Specifically, the authors present three crystal structures of PglH: a binary complex with the sugar donor UDP-GalNAc at 2.3Å, and two ternary complexes with a chemo-enzymatically generated LLO analog and either UDP or nonhydrolyzable UDP-CH₂-GalNAc, at 2.7 and 3.3Å, respectively. It is worth noting that the two ternary complexes represent the first crystal structures of a GT-B enzyme in the presence of lipid acceptor substrate/derivative. PglH displays a typical GT-B fold, consisting of two Rossmann-fold domains, with an important fissure in which UDP-GalNAc and the LLO analog are located. PglH displays a long amphipathic alpha helix in the N-terminal domain, which mediates membrane association. Interestingly, this alpha helix displays three positively charged residues that interact with the pyrophosphate group of the LLO acceptor substrate analog. In combination with enzymatic and molecular dynamics studies the authors propose a molecular ruler mechanism for glycan counting. Finally, the PglH/UDP-CH₂-GalNAc/LLO analog ternary complex suggest/support that PglH uses an SN₁-like mechanism. The article is clear and well written. Because of its novelty and the quality of the work, I strongly support the publication of this manuscript in Nature Communications.

Some questions to be addressed and suggestions for improvement are listed below:

1. page 2: *“Here, we present crystal structures of PglH in three distinct states”*. A good opportunity to describe that PglH is a retaining glycosyltransferase.

We added “retaining” to the abstract.

2. page 2: *“PglH catalyzes the transfer of exactly three N-acetylglalactosamine (GalNAc) units to the growing LLO precursor”*. Please, describe the sugar linkages between GalNAc residues transferred by PglH.

The description of sugar linkages has been added in the abstract.

3. page 3: *“Whereas PglA and PglJ each catalyzes a single GalNAc transfer reaction⁶, PglH catalyzes the addition of three GalNAc units”*. Please, describe the sugar linkages between GalNAc residues transferred by PglA, PglJ and PglH in the Introduction. Please add this information in Figure 1a and Supplementary Figure 1.

The description of sugar linkages has been added in the text and in both figures.

4. page 4: *“Although C. jejuni PglH is not an integral membrane protein, non-ionic detergents (Triton X-100 and dodecyl maltoside) were required for its successful purification in homogeneous and active form”*. What is the criteria to classify PglH as a non-integral membrane protein? Please add references accordingly. Any trace of protein in the soluble fraction?

We think the most widely accepted definition of an integral membrane protein describes a protein that spans the complete width of the membrane, with surface-exposed elements on both sides of the membrane. PglH is a membrane-associated protein.

5. page 18: Figure 2a. Please add a panel where the PglH-UDP-C-GalNAc is rotated 90 degrees – with the ruler helix perpendicular to the current view.

Two additional views of PglH have been introduced as Supplementary Figure 5.

- The complete amino acid sequence of the PglH construct should be included as a Supplementary Figure 2. In the case the authors used a synthetic pglH gene, with modified/improved DNA sequence, this information should be also included in Supplementary Figure 2.

The DNA sequence of the synthetic PglH gene, as well as the amino acid sequence of the construct used in this study, have been added to the Supplementary Information.

- Please include experimental phasing data in Table S1.

We have added this information in the methods section because the Nature Communications standardized table does not offer a place for this.

7. Spelling:

- “three N-acetylglalactosamine” by “three N-acetylgalactosamine”,

- “with the acceptor substrate UDP-GalNAc” by “with the donor substrate UDP-GalNAc”,

Corrected.

2. Reviewer #2:

The issue of polymerization control is of high interest in glycobiology. Nature makes polymers of defined length that would be hard to achieve by simple chemical means. The paper discusses a system for controlling lipid-linked oligosaccharide polymerization by limiting to three the number of UDP-GalNAc added to a trisaccharide. The paper presents the following experimental data,

1 The structure of the enzyme bound to UDP, UDP-CH2-GalNAc with LLO mimic (short lipid but trisaccharide)

2 The mutagenesis of a number of residues on the key helix (ruler).

3 Detailed modeling studies.

The structures reveal the active site and show the pyrophosphate of the LLO binds to R72 and K75. Mutagenesis shows that K75A, R72A slow the enzyme. The double mutant K75A, R72A is very slow. The most pertinent residue to their model for the ruler is K68. Mutation of K68 results in accumulation of pentasaccharide.

The thesis which is strongly supported by modeling is that the phosphate group jumps along the positively charged residues. In doing so it allows the substrate to grow by one residue at a time until the limit. In this model, the enzyme binds the tri, tetra and penta sugars (Figure 5).

This is interesting and novel. Further experimental support would strengthen the paper. The K68A mutant does not appear to affect 4 to 5 step, yet it is proposed to be important in Figure 5. Why not?

We apologize for the confusion that may have been caused by the drawing of Fig. 5. We have re-drawn the counting mechanism hypothesis (Fig. 5), now clearly showing that K68 is primarily involved in contacts associated with the third GalNAc transfer step. This is in agreement with our functional data where the K68A mutation caused an accumulation of the pentasaccharide LLO (i.e. the third GalNAc addition is slowed down).

What does the K68R look like?

For this study we decided to perform drastic mutations to highlight the role of the sidechains involved in the molecular ruler mechanism. However, further mutagenesis studies i.e. including K68R and others, which might help us to approach, in depth, the interactions of the molecular ruler, will be explored in further studies. In the absence of a detailed structural interpretation, we don't feel a K68R mutation would reveal anything meaningful, no matter what the outcome was. A structural study would require the large-scale generation of a pentasaccharide LLO analog and a new crystallization experiment, which we feel is outside of the scope of this manuscript.

It is also puzzling that since the K68 PP is proposed to be such a strong interaction (prevent competition of intermediates) why when it is abolished does the enzyme not go onto to create a 7 mer? If it creates the 6 mer without an anchor, why not longer?

The transfer of additional GalNAc units by wild type PglH to generate the heptasaccharide LLO is much less efficient than for the first three, since such a product could only be detected after long incubations of the reaction mix (16 hours). In the case of the mutant K68A, it would require even longer incubation times, which for *in vitro* assays would compromise the stability of the protein and hence be technically impossible.

In the model, the reducing sugar is pushed into a pocket in the tetra LLO. Have the authors considered trying to block the motion of the reducing sugar or otherwise block the re-arrangement of the LLO? Being able to identify such a mutant which would stall polymerization.

Neither of the two structures containing LLO revealed close contacts between PglH and the reducing end GlcNAc. We therefore wouldn't know what to mutate. However, for future studies we are considering mutating the surface of PglH aiming to find mutants that could lead to stalling of the glycan polymerization.

This is a good piece of work that deserves to be published in Nat Comm. If Nat Comm is to remain of the highest impact then given it's not the first ruler to be described (the Rock kinases should be cited) then pushing for that nit extra experimental validation is important. With that additional data, this paper will be seen as a landmark in disclosing a new mechanism of polymerization control.

Although we could not add the exact data proposed by this reviewer, we have added data of PglH acting on liposome-reconstituted LLO substrate, which strengthens the mechanistic proposal (see below).

3. Reviewer #3:

Ramirez et al report the synthesis of a soluble substrate analog and development of an enzymatic assay to measure the turnover of PglH glycosyltransferase in vitro. They determined

the structure of PglH, showing has an unusually long helix 2, with exposed basic residues. They determined this structure in the presence of the donor, as a product analog ternary complex, and as a reaction ready ternary complex inhibited by the presence of a non-reactive donor analog. Molecular dynamics simulations was used to explore the mode of interaction of PglH with the membrane, and to study possible interactions of the substrate with PglH. Finally, the authors mutate the three basic residues in helix 2 individually and in combination and show that these affect the relative accumulation rates of different products.

Overall the paper represents significant progress in our understanding of this enzyme in particular, and of the strategies that bacterial polysaccharide glycosyltransferases can employ to build specific glycostructures. A broad range of data are presented to support the paper's central ideas, with the approaches appropriate and the data appearing technically sound. The explication of the ternary complexes is a notable technical achievement, and the molecular ruler idea interesting, novel, and well presented. The methods are well enough documented to allow their replication by others in the field. The paper is well written, and argues its points clearly and succinctly. I believe that the implications are broad enough to interest the general readership of Nature Communications. My main quibble is that there is, in my view, some nuance missing around their data analysis and model of the molecular ruler hypothesis.

1) The reaction is carried out in the solution phase with a relatively soluble version of the substrate. While I fully appreciate the technical necessity, this alteration could reasonably result in behavior in this assay that does not reflect the behavior in cell. For example, the kinetics in solution reflect a three-dimensional search between acceptor and substrate, while the search in the membrane would be approximately two dimensional. In addition, in a lipid bilayer, phospholipids would compete with the substrate for charge-charge interactions with the basic side chains. And in solution, with no competing lipid tail interactions, the neryl-neryl group may interact with exposed hydrophobic patches on the protein, resulting in increased affinity and a possibly strong localization effect. I acknowledge that the ability of the solution assay to recapitulate the main outputs of the in vivo assay is a strong point in its favour, and the effect of the basic residue point mutations is strong, but I would still be more comfortable with this result if the authors were able to recapitulate the essential finding in vivo – e.g. by showing that the triple basic mutant fails to rescue a PglH knockout. At the very least, the limitations of interpreting the subtleties of a solution assay that stands in for a membrane embedded system should be clearly acknowledged in the discussion.

We agree and we have performed additional experiments in the form of activity measurements using wild-type PglH as well as variants containing mutation in the ruler helix and LLO reconstituted in liposomes. The results are comparable to the findings in detergents, although the accumulation of intermediates was reduced, suggesting that the processivity of the enzyme is higher when the experiment is performed in a lipid bilayer. The new results were added as Supplementary Figure 10 and discussed in the text.

2) Given that the mutation of the three basic residues is argued to affect the turnover of each of the three distinct substrates separately, ideally the data shown in fig 4b should be analyzed in terms of reaction rates with each distinct substrate (rather than aggregated into a single turnover number). Plotting the fraction of each substrate that has proceeded through a given step in the reaction as a function of enzyme concentration might allow the maximal rate of each step to be at least estimated for each construct. Quantifying the effect of each mutation on each step would allow a little more nuance in terms of discussing the mechanism of chain length

control. In particular, I suspect that all three residues have a measurable effect on all three steps of the reaction (e.g. by additional interactions with the GalNAc residues by R72/K75 in longer substrates). An alternative test of the idea that the sites are independent would be to test whether an equimolar mix of K75 protein plus K68 protein can act as effectively as the wild type.

Great suggestion. We have analyzed our data and have added a plot of the percentage of each LLO (substrate and products), as a function of the concentration of wild type PglH and the individual ruler helix mutants (Supplementary Figure 9). The findings support our mechanistic interpretations and are discussed in the text.

3) *The presentation of the mechanism of the molecular ruler is perhaps somewhat simplistic. For example, the basic side chains should have a fairly large degree of flexibility that would allow them to reorganize to accommodate different residue positions. Possibly a bidentate interaction with K68/Arg72 may still be possible, even with the GalNAc4 substrate. The text also in some places implies a one to one correspondence between the basic residues on helix 2 and the GalNAc residues added. E.g. Abstract “The ruler helix contains three positively charged side chains that can bind the pyrophosphate group of the LLO substrate and thus limit the addition of GalNAc units to three.”*

We have modified Fig. 5 (see also comment 1 of reviewer 2) to reduce confusion. We also modified the legend of Fig. 5 to indicate that the interactions of the pyrophosphate group of the LLO with residues of the ruler helix of PglH are hypothetical but are in line with our functional data. Importantly, the quantification of LLO intermediates during the reaction (see point 2 above) using the mutants K75A, R72A and K68A is in agreement with the proposed sequential role of these three residues of interacting with the pyrophosphate moiety.

4) *The PglH/UDP-CH2-GalNAc/LLO structure has very high average ADPs for the ligand in comparison to the rest of the structure. While the authors make clear that not all of this ligand is clear in the electron density map, it might be useful to discuss the limitations of interpreting this map in a little more detail. In particular, the pyrophosphate density from this map is the only direct evidence that the critical basic residues are required to bind these phosphate groups, so the quality of this evidence is a critical building block of the mechanism proposed.*

We agree with the reviewer that the average ADPs for the ligands in the structure containing UDP-CH2-GalNAc and LLO are high. However, that structure is not the only direct evidence of the interaction between the pyrophosphate group of the LLO and the ruler helix. In the structure of the ternary complex PglH–UDP–LLO, a better quality of the density was observed for both ligands, and the contacts between the pyrophosphate moiety and the side chains of K75 and R72 can be clearly deduced. Furthermore, our mutagenesis studies with residues of the ruler helix provide indirect evidence of their interaction with the LLO. We therefore don't think we're overstressing.

Additional points:

It is not accurate to refer to the PglH/LLO/UDP complex as a “product complex” (e.g. in the results). The LLO analog used has only 2 GalNAc residues added, and therefore cannot result from a PglH reaction. In addition the location of the LLO seems very similar to the positioning as required to act as an acceptor, rather than with the terminal GalNAc near where the GalNAc portion of UDP-GalNAc sits in the donor complex; a substrate with a single GalNAc (whose

reaction would result in this product) would presumably not be able to reach from the basic helix to the active site.

We agreed that this term could lead to confusion and have removed it.

Pg 7. “and ~3.4 Å from the methylene group” – I had to think for 5 seconds to figure out which methylene was being referred to. Maybe be more explicit – e.g. “the methylene group mimicking O4”.

Corrected

Pg 7 – Arg191 is argued to stabilize the leaving group on pg 5, with this idea being supported using the same the strong phenotype of a point mutation. While these experiments do not formally distinguish which function is the more critical in abrogating function, the wide conservation of this residue would argue that its role may not be primarily in binding the acceptor glycosyl group. I think it would be fairer to state that the importance of the Arg191 C2 acetamido interaction is not easily addressable by mutagenesis given its multiple functional roles.

We have inserted the following statement in Page 7: “However, because R191 also interacts with the pyrophosphate group of the leaving UDP, the R191A mutation has a dual effect.”

Fig 4a – In the figure caption, maybe indicate that this is the experimental structure, not an MD frame.

We have clarified in the figure caption that panel 4a corresponds to the crystal structure of the ternary complex PglH–UDP–LLO.

Fig 5 caption: Wording of the mechanism implies that the reaction is processive, with substrate being transferred between sites rather than being released between steps. No data is presented to prefer such a mechanism, and the accumulation of intermediates in even the wild type enzyme would seem to argue against it.

As mentioned by reviewer #1 and published earlier by Troutman and Imperiali (Biochemistry 2009, reference 7 in our paper), PglH is generally accepted to be a processive glycosyltransferase. In addition, the newly added functional data obtained using liposomes are in line with a processive mechanism, as virtually no intermediates (tetra- or penta-LLO) can be detected when wild type PglH is used.

Table 1 – it would be more useful to estimate the lower bound of activity detectable by the assay than state that no product was detected.

We have modified the legend to: “Turnover rate could not be calculated because no product could be detected or its level was below the detection limit of the assay.”

Methods - Pg 26. Is the described amino acid mix the stock concentration, or the final concentration. These values seem unreasonably high (total of 800 g / L total amino acids)

Well spotted! We have corrected the units to mg/L.

I understand that refining structures at low resolution is somewhat challenging, but the Ramachandran plot values for the PglH/UDP-CH2-GalNAc/LLO ternary complex seem a little poor, given that they are based on a well refined higher resolution binary structure.

We acknowledge that the Ramachandran plot values (1.68% outliers, 10.20% allowed and 88.13% favored) are not as high as for the other two structures, which were at higher resolution. However, we consider it somewhat artificial to manually fix residues to make the Ramachandran statistics look better. We therefore left things as they are and trust that readers will download all three structures for superposition.

Reviewers' Comments:

Reviewer #2:

Remarks to the Author:

The response addresses the critique.

Reviewer #3:

Remarks to the Author:

The authors have satisfactorily addressed all of the relatively minor points raised during review. I see no further impediments to publication.